# Phloretin Suppresses Bone Morphogenetic Protein-2-Induced Osteoblastogenesis and Mineralization via Inhibition of Phosphatidylinositol 3-kinases/Akt Pathway

**DOI:** 10.3390/ijms20102481

**Published:** 2019-05-20

**Authors:** Ayumu Takeno, Ippei Kanazawa, Ken-ichiro Tanaka, Masakazu Notsu, Toshitsugu Sugimoto

**Affiliations:** Internal Medicine 1, Shimane University Faculty of Medicine, 89-1, Enya-cho, Izumo, Shimane 693-8501, Japan; atakeno@med.shimane-u.ac.jp (A.T.); ken1nai@med.shimane-u.ac.jp (K.-i.T.); mnotsu25@med.shimane-u.ac.jp (M.N.); sugimoto@med.shimane-u.ac.jp (T.S.)

**Keywords:** phloretin, bone marrow stromal cell, ST2 cell, osteoblast, MC3T3-E1 cell, glucose uptake, osteoblastogenesis, adipogenesis, PI3K/Akt pathway

## Abstract

Phloretin has pleiotropic effects, including glucose transporter (GLUT) inhibition. We previously showed that phloretin promoted adipogenesis of bone marrow stromal cell (BMSC) line ST2 independently of GLUT1 inhibition. This study investigated the effect of phloretin on osteoblastogenesis of ST2 cells and osteoblastic MC3T3-E1 cells. Treatment with 10 to 100 µM phloretin suppressed mineralization and expression of osteoblast differentiation markers, such as alkaline phosphatase (ALP), osteocalcin (OCN), type 1 collagen, runt-related transcription factor 2 (Runx2), and osterix (Osx), while increased adipogenic markers, peroxisome proliferator-activated receptor γ (PPARγ), CCAAT/enhancer-binding protein α (C/EBPα), fatty acid-binding protein 4, and adiponectin. Phloretin also inhibited mineralization and decreased osteoblast differentiation markers of MC3T3-E1 cells. Phloretin suppressed phosphorylation of Akt in ST2 cells. In addition, treatment with a phosphatidylinositol 3-kinase (PI3K)/Akt inhibitor, LY294002, suppressed the mineralization and the expression of osteoblast differentiation markers other than ALP. GLUT1 silencing by siRNA did not affect mineralization, although it decreased the expression of OCN and increased the expression of ALP, Runx2, and Osx. The effects of GLUT1 silencing on osteoblast differentiation markers and mineralization were inconsistent with those of phloretin. Taken together, these findings suggest that phloretin suppressed osteoblastogenesis of ST2 and MC3T3-E1 cells by inhibiting the PI3K/Akt pathway, suggesting that the effects of phloretin may not be associated with glucose uptake inhibition.

## 1. Introduction

Bone is constantly remodeled by osteoclasts—bone resorbing cells, and osteoblasts—bone forming cells. Osteoblastic cells are derived from bone marrow stromal cells (BMSCs), which have the potential to differentiate into multicell lines, such as osteoblasts, adipocytes, chondrocytes, and myocytes [1]. BMSCs are thought to be involved in the development of bone fragility caused by pathological conditions, including aging [2,3,4,5], diabetes mellitus [5,6,7,8], estrogen deficiency [5], and glucocorticoid treatment [5,9,10]. Therefore, it is important to understand the mechanism of the commitment of BMSCs to osteoblasts or adipocytes.

Phloretin [2′,4′,6′-trihydroxy-3-(4-hydroxyphenyl)-propiophenone] is a natural phenolic compound which is contained in fruits, such as strawberry and apple [11,12]. It is reported that phloretin has pleiotropic effects, including anti-oxidative [13,14,15] and anti-inflammatory effects [16,17], as well as glucose uptake inhibition by inhibiting glucose transporters (GLUTs) [18,19,20,21,22]. Several studies have shown that phloretin promotes adipogenesis of preadipocytes [23,24]. In addition, we previously reported that phloretin promoted adipogenesis in mouse BMSC line ST2 [22], suggesting that phloretin may direct differentiation of BMSCs toward adipocytes. In contrast, a recent study reported the effect of phloretin on osteoblast differentiation. Antika et al. showed that oral administration of phloretin increased bone volume in an osteoporotic mouse model [25]. The authors also showed that treatment with phloretin enhanced mineralization of mouse osteoblast-like cell line MC3T3-E1, indicating that phloretin stimulates osteoblastogenesis. However, there are no reports to clarify the effect of phloretin on osteoblastogenesis in BMSCs thus far.

Glucose is a crucial nutrient for osteoblastogenesis. Guntur et al. reported that immature precursor cells prefer to use oxidative phosphorylation, and during differentiating to osteoblasts, the cells obtain marked preference for glycolysis to generate adenosine triphosphate [26]. Wei et al. and Li et al. showed that glucose uptake inhibition in osteoblasts suppressed osteoblastogenesis by proteosomal degradation of runt-related transcription factor 2 (Runx2), a master regulator of osteoblastogenesis [27,28]. These findings indicate that glucose uptake plays important roles in osteoblastogenesis. We previously reported that phloretin inhibited glucose uptake in ST2 cells [22]; therefore, it is speculated that phloretin may affect osteoblastogenesis mediated by glucose uptake inhibition.

The roles of phosphatidylinositol 3-kinase (PI3K)/Akt pathway in osteoblastogenesis [29,30,31,32,33,34,35] and adipogenesis [36,37,38,39,40] have been reported. Several studies reported that PI3K/Akt pathway was required for Runx2- and bone morphogenetic protein 2 (BMP-2)-induced osteoblastogenesis [29,30,31,34,35]. Furthermore, it has been shown that phloretin increased [24] or decreased [41] the phosphorylation of Akt. Therefore, it is considered that phloretin may affect osteoblastogenesis mediated by alteration of the PI3K/Akt pathway.

The present study aimed to investigate the effects of phloretin on BMP-2-induced osteoblast differentiation and mineralization in a mouse BMSC line ST2 and a mouse osteoblastic cell line MC3T3-E1 through PI3K/Akt pathway and glucose uptake inhibition. We also examined the alteration of adipogenic markers during BMP-2-induced osteoblastogenesis in those cells.

## 2. Results

### 2.1. The Effect of Phloretin on BMP-2-Induced Osteoblast Differentiation and Mineralization in ST2 Cells

We examined the effects of phloretin on osteoblast differentiation and mineralization of ST2 cells. Osteoblastogenesis was induced by 100 ng/mL BMP-2. von Kossa and Alizarin red stainings showed that 10–100 µM phloretin markedly suppressed the mineralization (Figure 1A). Next, we examined the expression of osteoblast differentiation markers. Treatment with phloretin significantly suppressed the mRNA expression of alkaline phosphatase (*Alp*) (Figure 1B,C), type 1 collagen (*Col-1*) (Figure 1D,E), and osteocalcin (*Ocn*) (Figure 1F,G), and crucial osteoblastogenic transcription factors, *Runx2* (Figure 1H,I) and osterix (*Osx*) (Figure 1J,K) in a dose-dependent manner on day 3 and day 5.

### 2.2. The Effect of Phloretin on Adipocyte Differentiation During BMP-2-Induced Osteoblastogenesis in ST2 Cells

We then investigated whether phloretin affected mRNA expression of adipogenic markers during BMP-2-induced osteoblastogenesis in ST2 cells. The cells were incubated in osteoblast differentiation medium with 0–100 µM phloretin, and mRNA expression of adipogenic markers, such as peroxisome proliferator-activated receptor γ (*Pparγ*), CCAAT/enhancer-binding protein α (*C/ebpα*), fatty acid synthase (*Fas*), fatty acid binding protein 4 (*Fabp4*), and adiponectin (*Apn*), was examined on day 3 and day 5 by real-time PCR. Phloretin significantly increased the expression of *Pparγ*, *Fabp4*, and *Apn* on day 3 (Figure 2A,G,I) and *Pparγ*, *C/ebpα*, *Fabp4*, and adiponectin *Apn* on day 5 (Figure 2B,D,H,J). The expression of *Fas* was not altered (Figure 2E,F).

### 2.3. The Effect of Phloretin on Mineralization and Expression of Osteoblastogenic and Adipogenic Markers in MC3T3-E1 Cells

We examined the effect of phloretin on osteoblast differentiation and mineralization in osteoblastic MC3T3-E1 cells. Osteoblastogenesis was induced by 100 ng/mL BMP-2 same as the examination using ST2 cells. von Kossa and Alizarin red stainings showed that treatment with 50 µM phloretin suppressed the mineralization (Figure 3A). Quantification of Alizarin red staining showed that the phloretin significantly inhibited the BMP-2-induced mineralization (Figure 3B). Then, we examined the expression of osteoblast differentiation markers. Phloretin at a concentration of 50 μM significantly suppressed the expression of *Alp*, *Col-1*, *Ocn*, and *Osx* (Figure 3C–E,G). The expression of *Runx2* was not altered by phloretin (Figure 3F). As to the expression of adipogenic markers, treatment with phloretin significantly decreased *C/ebpα* (Figure 3I). On the other hand, phloretin significantly increased *Fabp4* and *Apn* (Figure 3K,L). The expression of *Pparγ* tended to be increased, although the difference was not significant (Figure 3H). Taken together, it seems that phloretin increased adipogenic markers during BMP-2-induced osteoblastogenesis in MC3T3-E1 cells.

### 2.4. The Effect of Phloretin on Akt Phosphorylation and the Effects of a PI3K/Akt Inhibition on Osteoblastogenesis and Adipogenesis During BMP-2-Induced Osteoblastogenesis in ST2 Cells

The involvement of PI3K/Akt pathway in osteoblastogenesis and adipogenesis has been reported [29,30,31,32,33,34,35,36,37,38,39,40]. In the present study, we thus investigated the effects of phloretin on PI3K/Akt pathway in ST2 cells. After incubation in osteoblast differentiation medium for 2 days, the cells were treated with phloretin, and the phosphorylation of Akt was examined by western blotting. Phloretin at a concentration of 100 μM suppressed the phosphorylation of Akt after 1 h, and the phloretin-induced suppression of Akt lasted for 12 h (Figure 4A). Moreover, 10 to 100 μM phloretin significantly and dose-dependently inhibited the Akt phosphorylation (Figure 4B,C).

Next, we investigated the effects of inhibition of the PI3K/Akt signal on the mineralization and expression of osteoblast and adipocyte differentiation markers. Treatment with a 5 μM PI3K/Akt inhibitor LY294002 clearly decreased the mineralization (Figure 4D). Treatment with 2.5 to 10 μM LY294002 significantly suppressed the expression of *Col-1*, *Ocn*, *Runx2*, and *Osx* (Figure 4F–I) on day 3, although *Alp* expression was not affected (Figure 4E). These results suggested that phloretin-induced inhibition of osteoblast differentiation and mineralization except for ALP expression might be mediated by the suppression of the PI3K/Akt pathway.

With regard to adipogenic markers, treatment with 2.5 to 10 μM LY294002 significantly decreased the expression of *Fas*, *Fabp4*, and *Apn* (Figure 4L–N), which was opposite to the effect of phloretin on adipogenic markers (Figure 2G–J). By contrast, 10 μM LY294002 significantly increased *C/ebpα* (Figure 4K). The expression of *Pparγ* was not affected by LY294002 (Figure 4J).

### 2.5. The Effects of Glut1 Silencing on the Expression of Osteoblast and Adipocyte Differentiation Markers During BMP-2-Induced Osteoblastogenesis in ST2 Cells

We have shown that ST2 cells express only GLUT1 among a family of facilitative GLUT isoforms and that phloretin and *Glut1* silencing by siRNA inhibits glucose uptake in ST2 cells [22]. In the present study, we investigated the effect of *Glut1* silencing on osteoblastogenesis. Consistent with our previous study [22], silencing of *Glut1* sufficiently suppressed the *Glut1* expression (Figure 5A). Then, the effect of *Glut1* silencing on mineralization was examined. von Kossa and Alizarin red stainings showed that *Glut1* silencing did not affect the mineralization of ST2 cells (Figure 5B). Thereafter, the effects of *Glut1* silencing on the expression of osteoblast and adipocyte differentiation markers were examined. *Glut1* silencing significantly increased *Alp*, *Runx2*, and *Osx* (Figure 5C,F,G) and decreased *Ocn* (Figure 5E). The expression of *Col-1* was not affected (Figure 5D). *Glut1* silencing increased *Pparγ*, *Fas*, and *Apn* (Figure 5H,J,L) and decreased *Fabp4* (Figure 5K), whereas the expression of *C/ebpα* was not altered (Figure 5I). Taken together, the effects of *Glut1* silencing on the expression of osteoblast and adipocyte differentiation were inconsistent with the effects of phloretin.

## 3. Discussion

We previously reported the adipogenic effects of phloretin on bone marrow stromal ST2 cells [22]. However, there are no studies which investigate the effects of phloretin on osteoblastogenesis of BMSCs. In the present study, we showed that phloretin suppressed the osteoblast differentiation and mineralization in ST2 cells and MC3T3-E1 cells. In addition, phloretin increased the adipogenic markers even under the induction of osteoblastogenesis by BMP-2. Phloretin suppressed the Akt phosphorylation, and the PI3K/Akt inhibitor significantly inhibited the osteoblast differentiation and mineralization. These findings suggest that the suppression of the PI3K/Akt pathway might be involved in the inhibitory effects of phloretin on BMP-2-induced osteoblastogenesis. We previously demonstrated that phloretin and *Glut1* silencing decreased glucose uptake in ST2 cells [22]. However, silencing of *Glut1* had no effect on mineralization. Moreover, the effects of *Glut1* silencing on osteoblastogenic and adipogenic markers were inconsistent with those of phloretin, suggesting that the effects of phloretin on osteoblastogenesis are independent of glucose uptake inhibition.

Runx2 is a crucial transcription factor for osteoblastogenesis [42,43]. In the present study, treatment with phloretin suppressed the expression of Runx2 in ST2 cells but not in MC3T3-E1 cells. The effects of phloretin on Runx2 expression might be different in various stages of osteoblast differentiation. Moreover, the osteoblastogenic effect of Runx2 is regulated by various mechanisms other than transcriptional expression. It has been reported that phosphorylation of Runx2 enhances the osteoblastogenic activity of Runx2 [44,45,46]. Moreover, it has been reported that BMP-2-induced osteoblastogenesis is mediated by the interaction of Runx2 and Smad, a key component of BMP-2 signal [47,48]. Hence, there is a possibility that phloretin inhibited the BMP-2-induced osteoblastogenic activity of Runx2 by affecting the phosphorylation of Runx2 or the interaction of Runx2 with Smad. Further investigation is necessary to clarify this point.

There are several in vivo and in vitro studies which investigated the effects of phloretin on the bone. Lee et al. reported that oral administration of phloretin ameliorated bone loss through retarding bone resorption in ovariectomized mice [41,49]. Furthermore, Antika et al. reported that oral administration of phloretin increased bone volume, decreased serum receptor activator of nuclear factor-κB ligand/osteoprotegerin ratio, and diminished tartrate-resistant acid phosphate-positive osteoclasts in senescence-accelerated resistant mouse strain prone-6 mice, a senile mouse model [25]. In their study, treatment with 10 and 20 µM phloretin promoted the differentiation and mineralization of MC3T3-E1 [25]. These findings suggest that phloretin may increase bone volume via not only suppressing osteoclastogenesis but also promoting osteoblastogenesis, which is inconsistent with our results showing that phloretin decreased osteoblastogenesis in ST2 cells and MC3T3-E1 cells. With regard to this discrepancy, Antika et al. induced osteoblast differentiation of MC3T3-E1 cells by dexamethasone treatment. By contrast, we used BMP-2 for osteoblastogenesis of ST2 cells and MC3T3-E1 cells. Therefore, the discrepancy might be attributable to the difference between the methods of osteoblastogenic induction.

PI3K/Akt pathway is associated with both osteoblastogenesis [29,30,31,32,33,34,35] and adipogenesis [36,37,38,39,40]. It has been reported that PI3K/Akt pathway is required for BMP-2-induced osteoblastogenesis [29,30,31,32,33,34,35] via enhancing nuclear translocation of Smad1/5 [29], DNA binding of Runx2 [30], and binding of Smad1/5 to Osx promoter [35]. Moreover, forced expression of Runx2 increased the expression of PI3K subunits and Akt [30], suggesting that Runx2 signals and PI3K/Akt pathway may reciprocally affect each other. The present study demonstrated that phloretin inhibited the PI3K/Akt pathway and suppressed the osteoblastogenesis. As osteoblastogenesis was induced by BMP-2 in this study, it is assumed that phloretin suppressed the osteoblastogenesis by preventing the effect of BMP-2 via the suppression of PI3K/Akt signals. On the other hand, several studies showed the requirement of PI3K/Akt pathway for adipogenesis [36,37,38,39,40]. However, in this study, phloretin increased the adipocyte differentiation markers, such as *Fas*, *Fabp4*, and *Apn*, whereas LY294002, a PI3K/Akt inhibitor, decreased them, suggesting that inhibition of PI3K/Akt pathway is not associated with phloretin-induced upregulation of adipogenic markers.

Glucose is an important nutrient in osteoblastogenesis [26,27,28]. Wei et al. showed that glucose uptake inhibition suppressed the osteoblastogenesis by proteosomal degradation of Runx2 in osteoblast-specific GLUT1 knockout mice [27]. Li et al. also reported that knockout of Mst1/2 kinases suppressed the GLUT1 expression and glucose uptake, leading to the suppression of osteoblast differentiation and bone formation [28]. In addition, Karvande et al. showed that glucose uptake is required for parathyroid hormone-induced osteoblastogenesis by increasing glucose uptake [50]. These findings suggest that glucose uptake plays important roles in osteoblast differentiation. By contrast, unlike phloretin treatment, silencing of *Glut1* did not suppress mineralization, and the effects of *Glut1* silencing on the expression of osteoblastogenic markers were different from those of phloretin. These results suggest that the inhibitory effect of phloretin on osteoblastogenesis in ST2 cells is independent of glucose uptake inhibition, and that glucose uptake inhibition by *Glut1* silencing does not affect osteoblastogenesis in BMSCs. With regard to this discrepancy, Guntur et al. reported that immature precursor cells prefer to use oxidative phosphorylation to produce adenosine triphosphate, on the other hand, when they differentiated to osteoblasts, they gain strong preference for glycolysis [26]. These findings may suggest that osteoblasts mainly use glucose, whereas their precursor cells, that is, bone marrow stromal cells, can use not only glucose but also amino acids and fatty acids. Therefore, in our study, the effect of *Glut1* silencing might be compensated by other nutrients, such as amino acids or fatty acids. Another explanation is that the efficacy and duration of *Glut1* silencing might not be enough to examine the mineralization of ST2 cells. Thus, further investigation is necessary to elucidate the roles of glucose uptake for osteoblastogenesis in BMSCs.

In conclusion, the present study showed that phloretin suppressed the BMP-2-induced osteoblast differentiation and mineralization by inhibiting the PI3K/Akt pathway and that the effect of phloretin might be independent of glucose uptake inhibition.

## 4. Materials and Methods

### 4.1. Reagents

Cell culture medium and supplements were purchased from GIBCO-BRL (Rockville, MD, USA). Recombinant human BMP-2 was purchased from Peprotech (Offenbach, Germany). Phloretin, a PI3K/Akt inhibitor LY294002, and anti-β actin antibody were purchased from Sigma–Aldrich (St. Louis, MO, USA). Antibodies against phospho-Akt (Ser473) and total-Akt were purchased from Cell Signaling (Beverly, MA, USA).

### 4.2. Cell Cultures

Mouse BMSC line ST2 and mouse clonal osteoblastic cell line MC3T3-E1 were purchased from the RIKEN Cell Bank. ST2 cells and MC3T3-E1 cells were cultured in α-minimum essential medium (α-MEM) supplemented with 10% fetal bovine serum (FBS) and 1% penicillin-streptomycin in 5% CO_2_ at 37 °C. The medium was changed twice a week, and the cells were passaged when they were 80% confluent. For osteoblast differentiation and mineralization assay, ST2 cells were cultured in α-MEM supplemented with 10% FBS, 1% penicillin-streptomycin, 5 mM β-glycerophosphate, 100 mg/L ascorbic acid, and 100 ng/mL BMP-2 for 14 days after reaching confluence, and MC3T3-E1 cells were cultured in α-MEM supplemented with 10% FBS, 1% penicillin-streptomycin, 10 mM β-glycerophosphate, 50 mg/L ascorbic acid, and 100 ng/mL BMP-2 for 14 days after reaching confluence. The medium was changed twice a week.

### 4.3. Mineralization Stainings

The mineralization of ST2 cells and MC3T3-E1 cells was determined in 6-well plates or 24 well plates using von Kossa staining and Alizarin red staining. After incubation in osteoblast differentiation medium for 14 days, the cells were stained with 2% AgNO_3_ and fixed with 2.5% NaS_2_O_3_ by the von Kossa method to detect phosphate deposits in bone nodules [51]. At the same time, the order plates were fixed with ice-cold 70% ethanol and stained with Alizarin red to detect calcification [51]. For quantification, cells stained with Alizarin red (*n* ≥ 6) were destained with ethylpyridium chloride, then the extracted stain was transferred to a 96-well plate, and the absorbance at 595 nm was measured using a microplate reader.

### 4.4. Quantification of Gene Expression Using Real-Time Polymerase Chain Reaction (PCR)

Total RNA was extracted from the cultured ST2 cells and MC3T3-E1 cells using Trizol reagent (Invitrogen, San Diego, CA, USA) according to the manufacturer’s recommended protocol. Two micrograms of total RNA was transcribed into single-stranded cDNA (cDNA synthesis kit; Invitrogen, San Diego, CA, USA). Then, we used SYBR green chemistry to examine the mRNA expression of osteoblastogenic markers, *Alp*, *Col-I*, *Ocn*, *Runx2*, and *Osx*, and adipogenic markers, *Pparγ*, *C/ebpα*, *Fas*, *Fabp4*, and *Apn*. A housekeeping gene, *36b4*, was used to normalize the differences in the efficiencies of reverse transcription. The primer sequences are described in Table 1. Real-time PCR was performed using 50 ng of cDNA in a 25 μL reaction volume with Thermal Cycler Dice Real Time System Ⅱ (Takara Bio, Shiga, Japan). The double-stranded DNA-specific dye SYBR Green I was incorporated into the PCR buffer provided in the SYBR Green Real-time PCR Master Mix (Toyobo Co. Ltd., Tokyo, Japan) to enable quantitative detection of the PCR product. The PCR conditions were 95 °C for 15 min, 40 cycles of denaturation at 94°C for 15 s, and annealing and extension at 60°C for 1 min.

### 4.5. Western Blot Analysis

For western blot analysis, the cells were plated in 6-well plates and cultured as described above. After the cells were confluent, they were incubated in osteoblast differentiation medium for 2 days and treated with phloretin for up to 12 h. Then, the cells were rinsed with ice-cold phosphate-buffered saline (PBS) and scraped on ice into lysis buffer (BIO-RAD, Hercules, CA, USA) containing 65.8 mM Tris-HCl (pH 6.8), 26.3% (*w*/*v*) glycerol, 2.1% SDS, and 0.01% bromophenol blue to which 2-mercaptoethanol was added to achieve a final concentration of 5%. The cell lysates were sonicated for 20 s. The cell lysates were electrophoresed using 10% SDS-PAGE and transferred to a nitrocellulose membrane (BIO-RAD, Hercules, CA, USA). The blots were blocked with Tris-buffered saline (TBS) containing 1% Tween 20 (BIO-RAD, Hercules, CA, USA) and 3% bovine serum albumin (BSA) for 1 h at 4 °C. Thereafter, the blots were incubated overnight at 4 °C with gentle shaking with a primary antibody for phosphorylated Akt and total Akt at a dilution of 1:1000 and primary antibodies for β actin at 1:5000. These blots were extensively washed with TBS containing 1% Tween 20 and were further incubated with a 1:5000 dilution of horseradish peroxidase-coupled immunoglobulin G (IgG) of specified animal species (rabbit or mouse) matched to the primary antibodies in TBS for 30 min at 4 °C. The blots were then washed in TBS containing 1% Tween 20 three times, and the signal was visualized using an enhanced chemiluminescence technique. The bands were quantified with a software Image J [52]. The results were described as relative to control.

### 4.6. RNA Interference for GLUT1

RNA interference was used to downregulate the expression of Glut1 in ST2 cells. Small interfering RNA (siRNA) and SMARTpool reagents for GLUT1 and non-specific control siRNA duplexes were designed and synthesized by Dharmacon (Lafayette, CO, USA). For gene knockdown experiments, ST2 cells were seeded in 6-well plates and cultured at 37 °C for 48 h in α-MEM containing 10% FBS and antibiotics, followed by 24 h of incubation in medium without antibiotics. The cells were transfected with siRNAs (50 nM) by using DharmaFECT 1 Transfection reagent (Dharmacon, Lafayette, CO, USA) for 24 h according to the manufacturer’s instructions. Then, the cells were incubated in α-MEM supplemented with 10% FBS and antibiotics for another 24 h to reach confluence.

### 4.7. Statistics

Results are expressed as means ± standard error (SE). Statistical evaluations for differences between groups were performed using one-way analysis of variance followed by Fisher’s protected least significant difference. For all statistical tests, a value of *p* < 0.05 was considered a statistically significant difference.

## Figures and Tables

**Figure 1 ijms-20-02481-f001:**
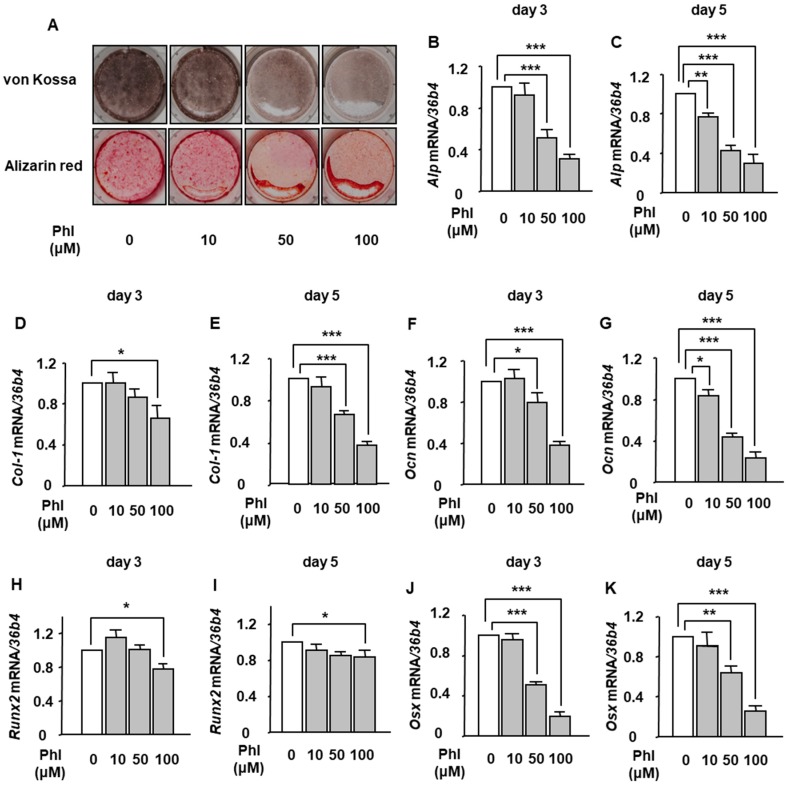
The effects of phloretin on osteoblast differentiation and mineralization in marrow stromal ST2 cells. (**A**) After reaching confluence, ST2 cells were incubated in osteoblast differentiation medium with 0, 10, 50, and 100 µM phloretin, and von Kossa staining and Alizarin red staining were performed at day 14. The results are representative of three different experiments. (**B**–**K**) After reaching confluence, the cells were incubated in osteoblast differentiation medium with 0–100 µM phloretin. The mRNA expression of osteoblast differentiation markers (*Alp*, *Col-1*, *Ocn*, *Runx2*, and *Osx*) was examined on day 3 and day 5 by real-time PCR. The results are expressed as mean ± SE (*n* ≥ 7). * *p* < 0.05, ** *p* < 0.01, *** *p* < 0.001. Phl; phloretin.

**Figure 2 ijms-20-02481-f002:**
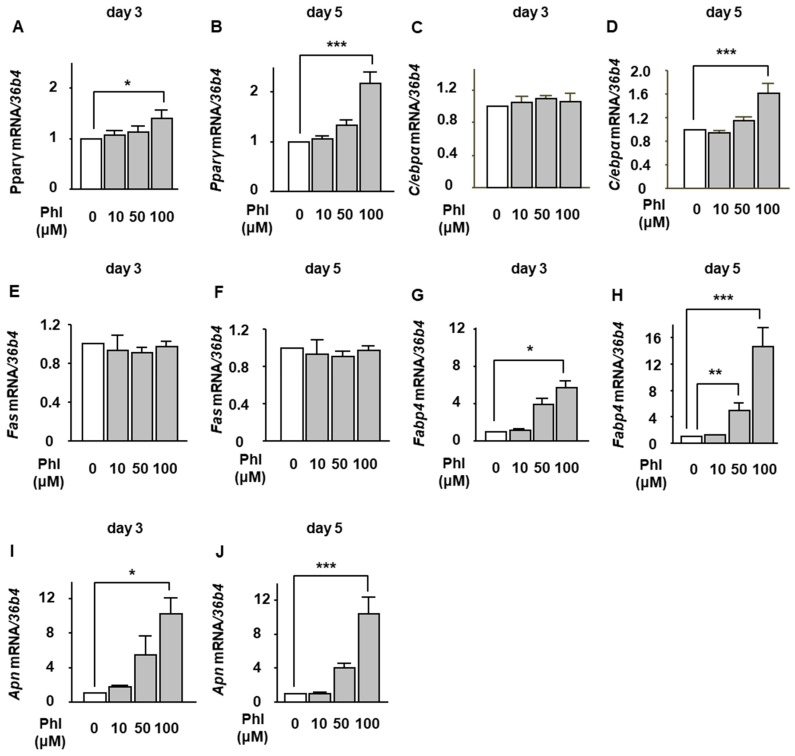
The effects of phloretin on the expression of adipocyte differentiation markers during BMP-2-induced osteoblastogenesis in ST2 cells. (**A**–**J**) ST2 cells were incubated in osteoblast differentiation medium with 0–100 µM phloretin, and the mRNA expression of adipogenic markers, *Pparγ*, *C/ebpα*, *Fas*, *Fabp4*, and *Apn*, was examined on day 3 and day 5 by real-time PCR. The results are expressed as mean ± SE (*n* ≥ 5). * *p* < 0.05, ** *p* < 0.01, *** *p* < 0.001. Phl; phloretin.

**Figure 3 ijms-20-02481-f003:**
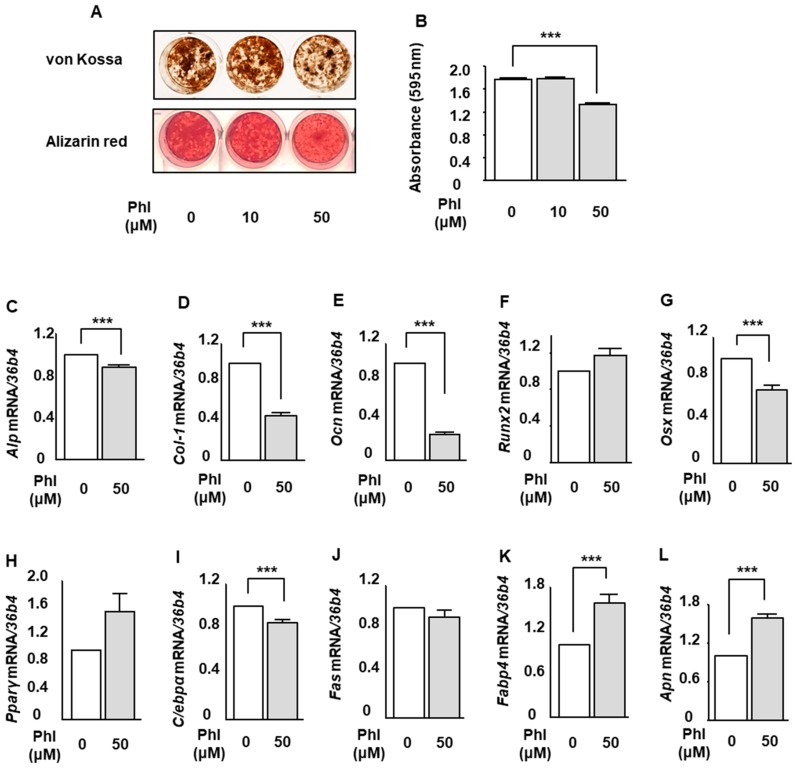
The effects of phloretin on mineralization and expression of osteoblastogenic and adipogenic markers in MC3T3-E1 cells. (**A**,**B**) After reaching confluence, MC3T3-E1 cells were incubated in osteoblast differentiation medium with 0, 10, and 50 µM phloretin, and von Kossa staining, Alizarin red staining, and its quantification were performed on day 14. Quantification results are expressed as mean ± SE (*n* = 6). *** *p* < 0.001. (**C**–**L**) After reaching confluence, MC3T3-E1 cells were incubated in osteoblast differentiation medium with 0 and 50 µM phloretin. The mRNA expression of osteoblast differentiation markers (*Alp*, *Col-1*, *Ocn*, *Runx2*, and *Osx*) and adipocyte differentiation markers (*Pparγ*, *C/ebpα*, *Fas*, *Fabp4*, and *Apn*) was examined on day 3 by real-time PCR. The results are expressed as mean ± SE (*n* ≥ 5). *** *p* < 0.001. Phl; phloretin.

**Figure 4 ijms-20-02481-f004:**
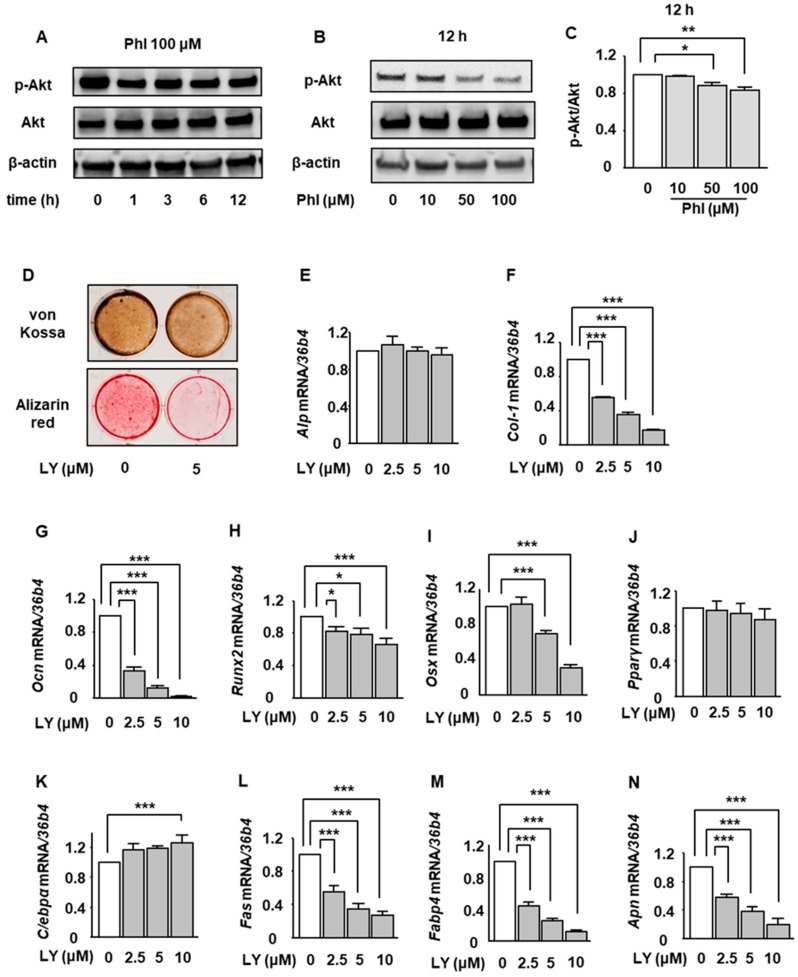
The involvement of suppression of PI3K/Akt pathway in the phloretin-induced downregulation of osteoblast differentiation markers and upregulation of adipocyte differentiation markers during BMP-2-induced osteoblastogenesis in ST2 cells. (**A**–**C**) After reaching confluence, ST2 cells were incubated in osteoblast differentiation medium for 2 days. Thereafter, the cells were treated with 100 µM phloretin for up to 12 h, total protein was extracted, and western blot analysis was performed to examine the time-dependent effect of phloretin on Akt (**A**). To test dose-dependency, the cells were treated with phloretin (0 to 100 µM) for 12 h (**B**). Quantification of the bands was performed (**C**). The results are representative of three experiments. Quantification results are expressed as mean ± SE (*n* = 3). * *p* < 0.05, ** *p* < 0.01. (**D**) After reaching confluence, ST2 cells were incubated in osteoblast differentiation medium with 0 or 5 µM LY294002, and von Kossa staining and Alizarin red staining were performed on day 14. (**E**–**N**) After reaching confluence, ST2 cells were incubated in osteoblast differentiation medium with 0 to 10 µM LY294002. Total mRNA was extracted on day 3, and the mRNA expression of osteoblastogenic markers (*Alp*, *Col-1*, *Ocn*, *Runx2*, and *Osx*) and adipogenic markers (*Pparg*, *C/ebpa*, *Fas*, *Fabp4*, and *Apn*) was examined by real-time PCR. The results are expressed as mean ± SE (*n* ≥ 5). * *p* < 0.05, ** *p* < 0.01, *** *p* < 0.001. Phl; phloretin, LY; LY294002.

**Figure 5 ijms-20-02481-f005:**
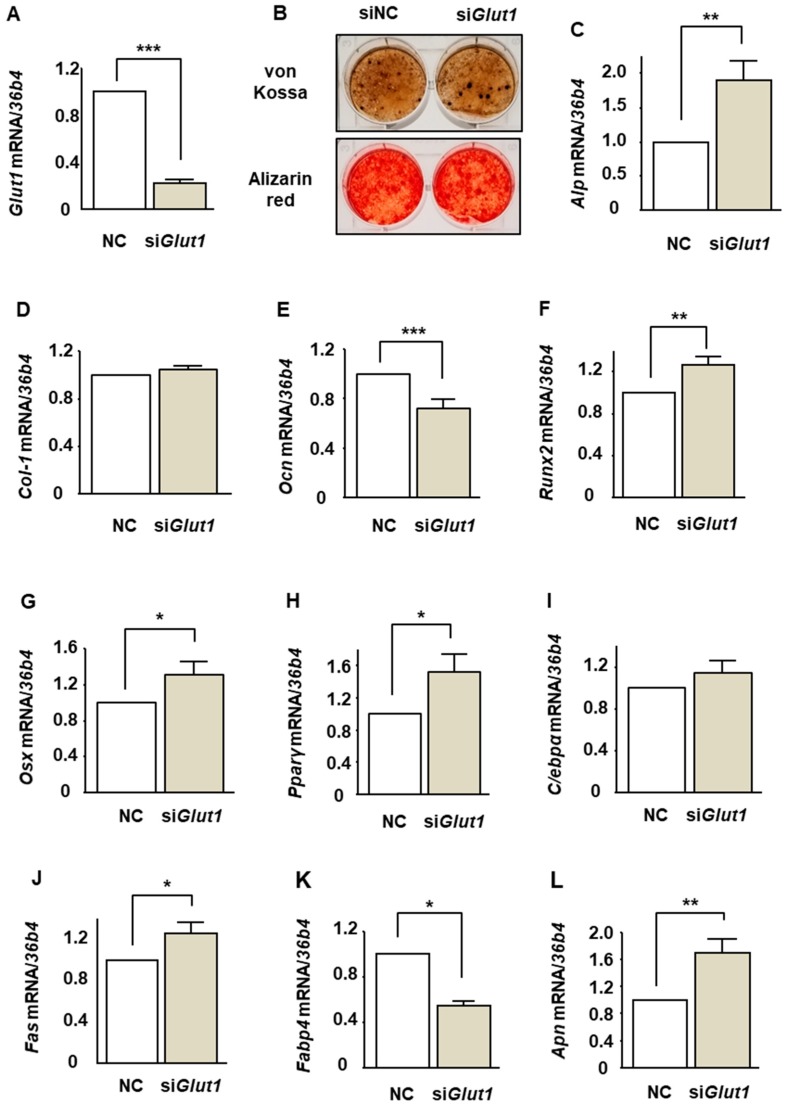
The effects of *Glut1* silencing on mineralization and the expression of osteoblast and adipocyte differentiation markers in ST2 cells. (**A**,**C**–**L**) After transfection of siRNA for *Glut1* and non-specific control, the cells were incubated in osteoblast differentiation medium for 3 days. Thereafter, the mRNA expression of osteoblastogenic markers (*Alp*, *Col-1*, *Ocn*, *Runx2*, and *Osx*) and adipogenic markers (*Pparg*, *C/ebpa*, *Fas*, *Fabp4*, and *Apn*) was examined by real-time PCR. *36b4* was used to normalize the differences in the efficiencies of reverse transcription. The results are expressed as mean ± SE (*n* ≥ 10). * *p* < 0.05, ** *p* < 0.01, *** *p* < 0.001. NC; negative control. (**B**) After transfection of siRNA, the cells were incubated in osteoblast differentiation medium, and the mineralization staining was performed on day 14.

**Table 1 ijms-20-02481-t001:** Gene names and their primers.

Gene Name	Primers	Accession No.
*36b4*	AAGCGCGTCCTGGCATTGTCT	NM_007475
	CCGCAGGGGCAGCAGTGGT	
*Alp*	CCGATGGCACACCTGCTT	X13409
	GAGGCATACGCCATCACATG	
*Col-1*	AACCCGAGGTATGCTTGATCT	NM_007742
	CAGTTCTTCATTGCATTGC	
*Ocn*	TGCTTGTGACGAGCTATCAG	L24431
	GAGGACAGGGAGGATCAAGT	
*Runx2*	AAGTGCGGTGCAAACTTTCT	NM_009820
	TCTCGGTGGCTGGTAGTGA	
*Osx*	CCCTTCTCAAGCACCAATGG	AF184902
	AGGGTGGGTAGTCATTTGCATAG	
*Pparg*	GTCTGTGGGGATAAAGCATC	NM_001127330.2
	CTGATGGCATTGTGAGACAT	
*C/ebpa*	TGAAGGAACTTGAAGCACA	NM_001287521.1
	TCAGAGCAAAACCAAAACAA	
*Fas*	CCCTTGATGAAGAGGGATCA	NM_007988.3
	ACTCCACAGGTGGGAACAAG	
*Fabp4*	TGGAAAGTCGACCACCATAAA	NM_024406.2
	GTCACGCCTTTCATGACACA	
*Apn*	TGTTGGAATGACAGGAGCTG	NM_009605.5
	TCCTTTTCACAAAGCCACACTAT	
*Glut1*	CGTCGTTGGCATCCTTAT	NM_011400.3
	TTCTTCAGCACACTCTTGG

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
