# Peer review of "Phloretin Suppresses Bone Morphogenetic Protein-2-Induced Osteoblastogenesis and Mineralization via Inhibition of Phosphatidylinositol 3-kinases/Akt Pathway"

_ijms, 2019, doi:10.3390/ijms20102481_

Round 1
Reviewer 1 Report
The authors have adequately addressed this reviewer's comments.
Author Response
We appreciate your kind review.
Reviewer 2 Report
The authors have made many requested changes to their initial paper. However, I’m still puzzled by several discrepancies with literature that the authors clearly recognize but do not have any rational explanation for them. My remaining concerns are outlined below:
1- In MC3T3 cells (Figure 3), phloretin increases a bit or had no effect of RUNX2 expression, the master gene of osteoblastogenesis. How could the expression of other osteoblastogenic markers, that are downstream RUNX2 expression, be decreased? The expression of these markers directly depend on that of RUNX2.
2- Response to comment 3: if the authors believe that discrepancy is due to the method of differentiation, this is easy to check for by using dexamethasone on their cells.
3- Minor: line 127, change 500 uM to 50 uM
Author Response
The reviewer’s comments
The authors have made many requested changes to their initial paper. However, I’m still puzzled by several discrepancies with literature that the authors clearly recognize but do not have any rational explanation for them. My remaining concerns are outlined below:
Comment 1
1- In MC3T3 cells (Figure 3), phloretin increases a bit or had no effect of RUNX2 expression, the master gene of osteoblastogenesis. How could the expression of other osteoblastogenic markers, that are downstream RUNX2 expression, be decreased? The expression of these markers directly depends on that of RUNX2.
Response to the comment 1
We appreciate your important comment. As you pointed out, phloretin did not affect the expression of Runx2 mRNA in the study. ST2 cells are multipotent stromal cells, and MC3T3-E1s are preosteoblasts. As Runx2 is a master regulator of osteoblastogenesis, it plays important roles in the early stage of osteoblastogenesis. Thus, the effects of phloretin on Runx2 expression might be different in ST2 and MC3T3-E1 cells. Moreover, it was reported that phosphorylation of Runx2 and interaction of Runx2 with Smad are also important for osteoblastogenic activity of Runx2. Therefore, there is a possibility that phloretin inhibited osteoblastogenesis by affecting phosphorylation of Runx2 or the interaction of Runx2 with Smad. We described about these in line 205-214. Again, thank you for the useful comment.
Comment 2
2- Response to comment 3: if the authors believe that discrepancy is due to the method of differentiation, this is easy to check for by using dexamethasone on their cells.
Response to the comment 2
Thank you very much for the appropriate comment. The effect of dexamethasone on osteoblastogenesis is known to be different between in vitro and in vivo, it is sometimes difficult to interpret experimental data. In this study, we focused on the effect of phloretin on BMP-2-induced osteoblastogenesis as we described in the introduction. Therefore, we used BMP-2 for induction of osteoblastogenesis of both ST2 cells and MC3T3-E1 cells. Moreover, the effect of phloretin on dexamethasone-induced mineralization in MC3T3-E1 cells has already been reported, we discussed our results with the previous reports.
Comment 3
3- Minor: line 127, change 500 uM to 50 uM
Response to the comment 3
Thank you for pointing out our careless mistake. We corrected it (line 126) and carefully rechecked the manuscript.
Round 2
Reviewer 2 Report
NA
This manuscript is a resubmission of an earlier submission. The following is a list of the peer review reports and author responses from that submission.
Round 1
Reviewer 1 Report
The manuscript by Takeno et al. investigates the effect of Phloretin on osteoblastogenesis. The authors propose that differentiation of the ST2 cell line used as a model is blocked by Phloretin through inhibition of the PI3K/Akt signaling pathway. The data and the interpretations made in the manuscript raise significant concerns.
1/ The authors published recently that phloretin favors differentiation of ST2 cells into adipocytes. It seems therefore relatively evident that if adipogenesis is favored, osteoblastogenesis will be impaired. Even if the differentiation media are different between the studies, phloretin may play a role upstream of the differentiation factors used. It would be important to do the same experiment on ST2 without differentiation medium, as a control.
2/ The authors claim that phloretin activates AMPK phosphorylation in ST2 cells but that AMPK activation does not account for the down-modulation of osteoblastic factors (and up-regulation of adipogenic factors as already shown in the previous study). In order to confirm this assumption, it must be shown by western blot that the AMPK inhibitor ara-A effectively inhibits the AMPK signaling pathway.
3/ It is not sufficient to discuss the discrepancy between this study and previous publications concerning the effect of phloretin, glucose uptake and AMPK activation on osteogenesis. This work on a single cell line only adds confusion to the field without giving concrete answers. It is essential to use the same conditions on at least another cell line and primary mesenchymal cells to prove that the effects are not cell line dependent.
Reviewer 2 Report
This study investigated the effect of phloretin on osteoblastogenesis of bone marrow stromal cell line ST2 cells. Treatment with phloretin suppressed mineralization and expression of alkaline phosphatase, osteocalcin, type 1 collagen, Runx2 and osterix, while increased adipogenic markers, PPARgamma , C/EBPα, fatty acid binding protein 4 and adiponectin. Furthermore this study explored the signaling pathways that are affected by phloretin by using pharmacological inhibitors for AMPK and PI3K/Akt, and oligonucleotide knockdown of GLUT1. Treatment with AMPK inhibitor, ara-A, prevented the phloretin-induced downregulation of OCN, whereas other osteoblast differentiation markers were not affected. Treatment with a PI3K/Akt inhibitor LY294002, suppressed mineralization and the expression of osteoblast differentiation markers other than ALP. GLUT1 silencing by siRNA did not affect mineralization but decreased OCN expression and increased ALP, Runx2 and Osx expression. The effects of GLUT1 silencing on osteoblast differentiation markers and mineralization were different to those of phloretin. The authors suggest that phlerotin suppressed osteoblastogenesis in ST2 cells by was due to inhibition of PI3K/Akt pathway and the effects of phloretin were independent of glucose uptake inhibition.
Minor comments:
1. Correct vocabulary throughout the manuscript i.e. page 1, line 21 dipsogenic markers
2. Do the osteogenic markers fluctuate over time? Are 3 days representative of gene expression?
3. What is the mineralization for Phloretin and ara-A treatments in Fig. 3?
4. What are the vehicles used for phloretin, LY294002, and ara-A? Are the vehicle controls for the different doses taken into account?
5. There are many osteogenic signaling pathways in addition to the ones explored by this study. Why are these pathways chosen specifically?
Major comments:
1. The study investigated the independent role of AMPK, PI3K/Akt and Glut1 signaling during phloretin suppression of osteoblastogenesis. It seems this study is a mix of 3 independent signaling pathways which loses focus. The experiments with AMPK and Glut1 opens up more questions than answers. The reviewer suggests a more focused study with more details on one experiment for this manuscript.
2. Phloretin inhibits osteoblastogenesis but phosphorylates AMPK at Thr172 (Fig. 3A-C). Since this is suggested to be due to lack of glucose uptake, experiments with overexpression of Glut1 in the presence of Phloretin may elucidate this phenomena.
3. Activation of PI3K/AKT signaling could be performed in the presence of Phloretin to elucidate the role of PI3K/AKT during phloretin suppression.
Reviewer 3 Report
Takeno and colleagues describe the impact of phloretin on a mouse bone marrow stromal cell line ST2.
Using conventional techniques, they showed that phloretin application interferes with mineralization in vitro when ST2cells undergo an osteoblastic differentiation. This process seems to be mediated by controlling the 3-kinases/Akt pathway. Phloretin-mediated osteogenesis inhibition is independent of the glucose uptake inhibition in the same cell line.
The authors also investigated the effect of phloretin on adipocyte differentiation during osteoblastogenesis using the same cell line.
In both differentiation processes, the authors showed used protein and transcript analyses, and pharmacology to draw their conclusions.
The experiments were well designed and data were presented with sufficient replicates for statistics, but the choice for some representative figures does not always follow the averaged graphs.
Major comments:
The data generated from this study and the conclusions drawn raise many questions:
The authors have already published another paper with almost the same title in this journal “Phloretin Promotes Adipogenesis via Mitogen-Activated Protein Kinase Pathways in Mouse Marrow Stromal ST2 Cells” (IJMS doi: 10.3390/ijms19061772)” using similar approaches. The used adipogenic medium to induce adipogenic differentiation in ST2 cells. In the actual paper, they looked again at the adipogenic differentiation but using osteogenic inducing medium. The osteogenic medium seems to differentiate the ST2 cells into osteoblasts and adipocytes at the same time. The data presented here do not show, neither state if these cells are adipocytes and can be oil red positive. Does the osteogenic medium produce two cell types in the dish? If so, what is the impact of this hypothesis on all osteogenic data generated. What is the proportions of the two cell type in each dish? How can one distinguish protein or transcript variation in an osteoblast or an adipocyte?
An oil red staining of the differentiated cells, in osteogenic inducing medium, would help to assess how heterogeneous the cultures are.
- Osteoblastogenesis/glucose uptake relationship:
In discussion, the authors pointed to many other studies showing, in vivo, that there’s a clear relation between glucose uptake and osteoblastogenesis. The authors stat in the title of the paper that this is not true without providing evidences that support this. The discussion provided does not help to explain the discrepancy of their data from the literature.
- Discrepancy ST2 versus in vivo and other cell type differentiation:
In vivo and in vitro models that have investigated the effect of phloretin on osteoblastogenesis seem to contradict the data generated on ST2 cells. The authors hypothesize that this may be due to culture conditions and/or cell type used, but this leaves the reader with strong doubts about the real and/or physiological role of Phloretin during osteoblastogenesis. The authors should provide evidences that ST2 cell line has a different behavior upon osteogenic differentiation by using another cell line such as the MC3T3-E1. In parallel, and as suggested by the authors, ST2 induction with dexamethasone should be used to evaluate the effect of phloretin on osteogenesis.